

# Hybrid deep learning approach for brain tumor classification using EfficientNetB0 and novel quantum genetic algorithm

Kerem Gencer[1] and Gülcan Gencer[2]

[1] Afyon Kocatepe University, Faculty of Engineering, Department of Computer Engineering, Afyonkarahisar, Turkey
[2] Afyonkarahisar Health Sciences University, Faculty of Medicine, Department of Biostatistics and Medical Informatics, Afyonkarahisar, Turkey

## ABSTRACT

One of the most complex and life-threatening pathologies of the central nervous system is brain tumors. Correct diagnosis of these tumors plays an important role in determining the treatment plans of patients. Traditional classification methods often rely on manual assessments, which can be prone to error. Therefore, multiple classification of brain tumors has gained significant interest in recent years in both the medical and computer science fields. The use of artificial intelligence and machine learning, especially in the automatic classification of brain tumors, is increasing significantly. Deep learning models can achieve high accuracy when trained on datasets in diagnosis and classification. This study examined deep learning-based approaches for automatic multi-class classification of brain tumors, and a new approach combining deep learning and quantum genetic algorithms (QGA) was proposed. The powerful feature extraction ability of the pre-trained EfficientNetB0 was utilized and combined with this quantum genetic algorithms, a new approach was proposed. It is aimed to develop the feature selection method. With this hybrid method, high reliability and accuracy in brain tumor classification was achieved. The proposed model achieved high accuracy of 98.36% and 98.25%, respectively, with different data sets and significantly outperformed traditional methods. As a result, the proposed method offers a robust and scalable solution that will help classify brain tumors in early and accurate diagnosis and contribute to the field of medical imaging with patient outcomes.

# INTRODUCTION

One of the most complex and life-threatening pathologies of the central nervous system is brain tumors. Timely accurate diagnosis and classification of these tumors plays a critical role in the treatment of patients. In traditional classifications, this process is time-consuming and prone to errors. Therefore, studies on automatic multiple classification of brain tumors arouse great interest. In particular, deep learning methods show promising results in automatically classifying brain tumors (*Litjens et al., 2017*). Deep learning models trained on large data sets have the ability to diagnose and classify with high

Corresponding author
Kerem Gencer,
keremgencer09@hotmail.com

accuracy rates (*Esteva et al., 2021*). Thus, brain tumors are automatically classified and diseases can be diagnosed more quickly and accurately (*Ghaffari, Sowmya & Oliver, 2019*). One of the most commonly used methods for classifying brain tumors is magnetic resonance imaging. Magnetic resonance imaging method allows a detailed examination of morphology and anatomical features by providing high-resolution and contrast images (*Mazurowski et al., 2019*). For this reason, it is known to be used effectively in training deep learning models. Figure 1 shows a plan consisting of brain tumor types and normal brain images. On the other hand, the development of automatic classification systems is difficult due to the availability of sufficient and balanced data sets. For this reason, decreases in model performance are observed in many studies (*Bakas et al., 2018*).

Methods such as data augmentation techniques and transfer learning are used to help use data sets effectively and increase the generalization capacity of models (*Shorten & Khoshgoftaar, 2019*). In the literature, deep learning models and different techniques have been used to classify brain tumors and perform feature selection. However, no study is known that integrates the EfficientNetB0 model using the quantum genetic algorithm and applies this method for feature selection. This combination makes our proposed approach unique with high accuracy. A combination of deep learning and quantum genetic algorithms was used in this study, creating an advanced and effective methodology for classifying brain tumors. The powerful feature extraction capabilities of the EfficientNetB0 model pre-trained in ImageNet were utilized and developed with a new feature selection method based on quantum genetic algorithms. The proposed hybrid approach addresses critical challenges in medical imaging, reducing the need for high-dimensional feature space and improving classification performance on limited, unbalanced datasets. It offers a robust and scalable solution, aiding in early and accurate diagnoses, thus advancing medical imaging research.

Key contributions of our study include:

- Two different brain tumor datasets were used to demonstrate the generalization ability of the proposed hybrid model and its performance on different data sources.
- As a result of comparing the proposed method (EfficientNetB0 Model with QGA-FS) with CNN, EfficientNetB0, CNN+EfficientNetB0, fine-tuning CNN+EfficientNetB0 models, it was possible to determine the model that gives the best results.
- The proposed hybrid model reached the highest accuracy rate in both data sets, revealing that using the quantum genetic algorithm for feature selection will increase performance. With feature selection made in a hybrid way, the most important features were determined and the model worked more efficiently and accurately. This made a significant contribution to the integration of the quantum genetic algorithm with deep learning models.
- Comparing the performance of different models in the study helped determine which model configuration was most effective in the brain tumor classification problem. This provides an important reference point for future research and practice.

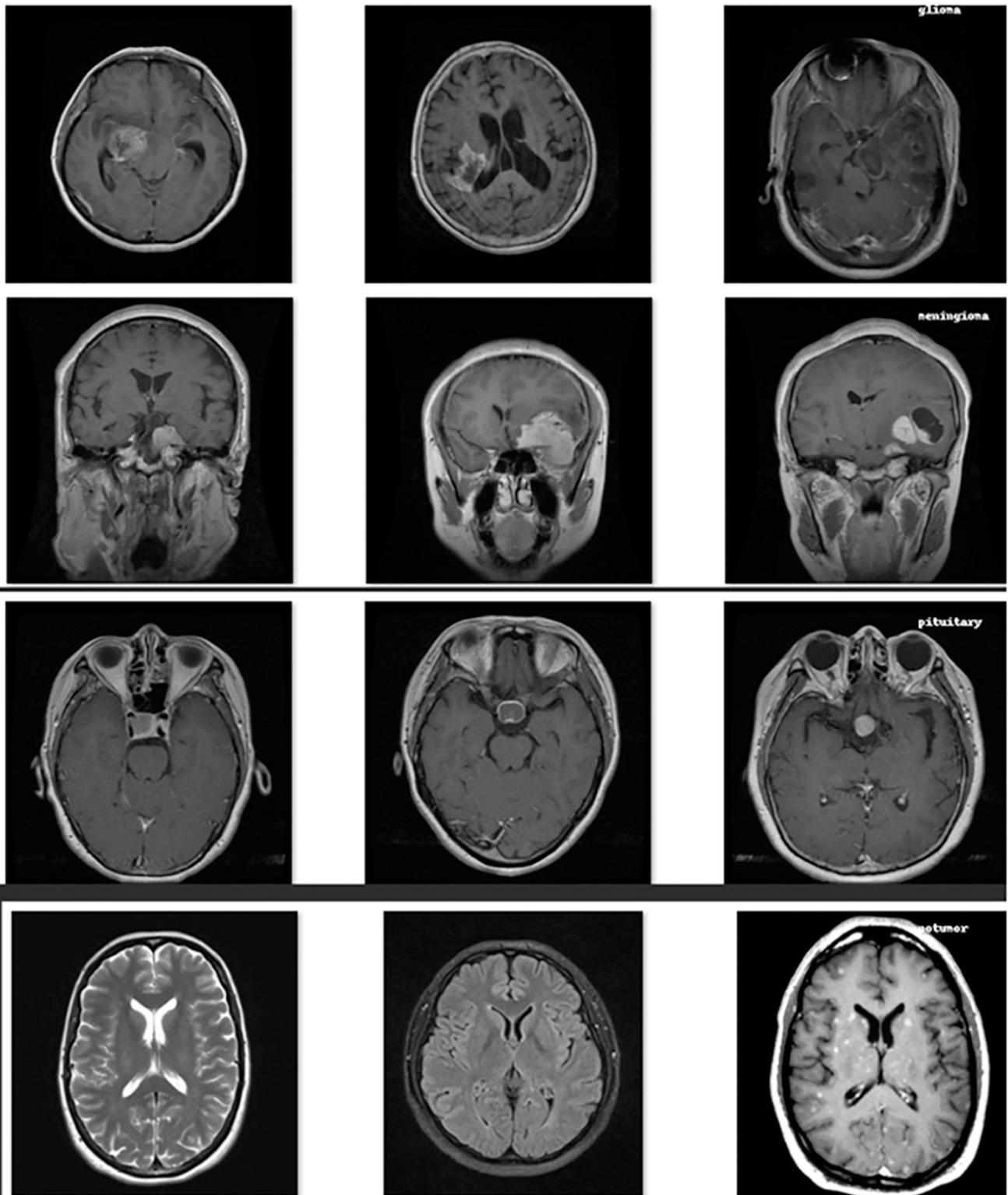

**Figure 1 Visualization of different brain tumor types and normal brain.**

These contributions show that the study both brings innovation to existing methods in the literature and aims to achieve higher accuracy rates in brain tumor classification. We believe that the use of quantum genetic algorithm (QGA) in our study has significantly enhanced the uniqueness and quality of our hybrid model. QGA offers a more effective feature selection and optimization compared to traditional methods, especially in the classification of biomedical images. This algorithm combines the power of quantum computing with genetic algorithms to produce faster and more efficient solutions, while also enabling the discovery of more complex relationships in large datasets. The use of QGA in our study not only allowed us to achieve high accuracy rates but also improved the overall performance of the model, presenting an innovative approach to classification tasks. As a result, we can confidently state that the proposed hybrid model distinguishes itself from similar works in the literature and provides a unique contribution.

## RELATED WORK

Classification of brain tumors has gained significant attention in recent years with the use of artificial intelligence (AI) and machine learning (ML) techniques in the field of medical imaging. Traditional methods are time-consuming and error-prone as they often rely on manual evaluations. Therefore, studies on automatic multiclass classification of brain tumors arouse great interest both in the medical field and in computer science.

*Bakas et al. (2017)* extended their Glioma MRI collection with expert segmentation labels and radiomic features. The Cancer Genome Atlas (TCGA) has improved the quality of the glioma MRI dataset, providing researchers with richer and more detailed data sets, and has become an important resource in the development of glioma diagnosis and classification models. *Cheng et al. (2015)* used tumor area augmentation and division methods to increase performance in brain tumor classification, and classification accuracy was increased by providing more detailed and precise identification of tumors. The study provided significant gains, especially when working with small and unbalanced data sets. *Esteva et al. (2021)* performed skin cancer classification at the dermatologist level using deep learning networks and showed that deep learning models trained on large data sets can diagnose with accuracy comparable to human experts. *Ghaffari, Sowmya & Oliver (2019)* performed brain tumor segmentation using multimodal brain scans and three-dimensional convolutional neural networks, and a more accurate segmentation of tumors was achieved by integrating data obtained from different imaging modalities. *Litjens et al. (2017)* presented a comprehensive review on the use of deep learning techniques in radiology and provided a broad perspective on how deep learning models can be used in radiological image analysis and became an important reference source for researchers. *Mazurowski et al. (2019)* presented an overview of the concepts and current status of deep learning in radiology and examined in detail the potential and application areas of deep learning techniques in radiological image analysis.

*Pereira et al. (2016)* used convolutional neural networks for glioma segmentation and demonstrated the effectiveness of these models in distinguishing different tumor types.

Thus, it highlights the success of deep learning models in automatic segmentation of complex and heterogeneous tumors such as glioma. *Shorten & Khoshgoftaar (2019)* present a survey on image data augmentation techniques for deep learning, detailing how data augmentation methods can improve the performance of deep learning models and the applicability of these techniques. *Sultan, Salem & Al-Atabany (2019)* performed multi-class brain tumor image classification using a deep neural network. Deep neural networks were used to accurately classify different brain tumor types and high accuracy rates were achieved. *Akhtar & Mian (2018)* presented a survey on the threats of adversarial attacks to deep learning in computer vision. This study examines the vulnerabilities of deep learning models and their defense strategies against hostile attacks. In *Zhang et al. (2015)*, deep convolutional neural networks were used for multimodal isointense infant brain image segmentation. This study demonstrates the effectiveness of deep learning techniques in segmenting infant brain images. In *Afshar, Plataniotis & Mohammadi (2019)*, brain tumor classification based on MRI images was performed using capsule networks. Capsule networks have been particularly effective in learning complex data structures and improving model performance. *Liu et al. (2023)* presented a survey on deep learning for brain tumor segmentation. This study comprehensively reviews deep learning techniques in the field of brain tumor segmentation and current advances in this field. *Abd-Ellah et al. (2019)* presented a review on brain tumor diagnosis from MRI images and examines the different techniques used in the analysis of MRI images and how these techniques affect diagnostic accuracy. *Kamnitsas et al. (2017)* used efficient multiscale 3D CNN with fully connected CRF for brain lesion segmentation, demonstrating the effectiveness of multiscale 3D CNNs in segmentation of brain lesions. *Roy et al. (2019)* developed QuickNAT, a fully convolutional network for fast and accurate segmentation of neuroanatomy. QuickNAT provided high accuracy and speed in neuroanatomy segmentation. *Wang et al. (2019)* made an aleatoric uncertainty estimation with test time increase for medical image segmentation and examined how uncertainty estimation techniques can increase model reliability. *Havaei et al. (2017)* performed brain tumor segmentation with deep neural networks and evaluated the performance of deep neural networks in brain tumor segmentation. *Milletari, Navab & Ahmadi (2016)* developed V-net, a fully convolutional neural network for volumetric medical image segmentation, and V-net showed high performance in the segmentation of 3D medical images. *Zhao et al. (2018)* developed a deep learning model integrating FCNNs and CRFs for brain tumor segmentation, providing high accuracy in segmenting different tumor types. *Akkus et al. (2017)* presented a review on deep learning for brain MRI segmentation and comprehensively examined the deep learning techniques used in the field of brain MRI segmentation. *Hosseini-Asl, Keynton & El-Baz (2016)* diagnosed Alzheimer's disease with the adaptation of a 3D convolutional network and demonstrated the use of deep learning techniques in the diagnosis of Alzheimer's disease. *Beers et al. (2021)* developed DeepNeuro, an open-source deep learning toolbox for neuroimaging. DeepNeuro facilitates the application of deep learning techniques in neuroimaging analysis.

## METHODS

### CNN-based model

In our study, a traditional convolutional neural network (CNN) model was developed for brain tumor classification. The model consists of several layers, including convolutional, max pooling, and fully connected layers, designed to extract features and perform classification tasks. The CNN accepts 128 × 128 pixel RGB brain tumor images as input. It uses two convolutional layers (32 and 64 filters) with ReLU activation, followed by max pooling layers to reduce the image size and retain important features. The extracted features are flattened and passed through a fully connected layer with 128 neurons, followed by a softmax layer for classification into three tumor types (glioma, meningioma, pituitary tumor). To prevent overfitting, dropout with a rate of 50% was applied. The model was trained using the sparse categorical cross-entropy loss function and the Adam optimizer, with early stopping and learning rate reduction techniques employed to enhance performance. The dataset was split 80% for training and 20% for testing, and the model was trained for up to 50 epochs, with early stopping applied to halt training when no improvement was observed (*Pereira et al., 2016*).

### EfficientNetB0 based model

In this study, the EfficientNetB0 model was utilized with transfer learning for brain tumor classification. EfficientNetB0, a deep learning model pre-trained on the ImageNet dataset, was fine-tuned for the specific task of classifying brain tumors. By leveraging pre-trained weights, the training time was reduced, and performance was improved. The model's pre-trained layers were frozen, except for the final layers, which were modified for the classification task. The output features were flattened and passed through a dense layer with 128 neurons, followed by a dropout layer to prevent overfitting. The final softmax layer classified the images into glioma, meningioma, and pituitary tumor categories. The model was trained using the Adam optimizer and sparse categorical cross-entropy loss function, with early stopping and learning rate reduction applied for optimization. The EfficientNetB0 model demonstrated high accuracy in brain tumor classification (*Pereira et al., 2016*).

### Hybrid model

In this study, a hybrid model combining CNN and EfficientNetB0 was developed to improve brain tumor classification accuracy. The hybrid model leverages CNN's ability to extract low-level features and EfficientNetB0's strength in extracting high-level features. The features from both models are concatenated and passed through dense layers for classification. This combination enhances performance by utilizing the strengths of both models. The hybrid model was trained with early stopping and learning rate reduction techniques, preventing overfitting. After training, the model was evaluated on a test set, showing high accuracy and superior performance in brain tumor classification (*Pereira et al., 2016*).

## Fine-tuned hybrid model

In this study, the performance of the hybrid model combining CNN and EfficientNetB0 was further enhanced through the fine-tuning method. Fine-tuning is an optimization process where certain layers of a pre-trained model, such as EfficientNetB0, are retrained to better adapt to a new dataset. During the transfer learning phase, the core layers of the EfficientNetB0 model were frozen and not trained. However, in the fine-tuning phase, the final layers of the model were unlocked and retrained to better fit the new dataset. The last few layers of EfficientNetB0 were adapted specifically for the brain tumor classification task, allowing the model to learn more distinct features and improve its overall performance. During training, only the last layers of the EfficientNetB0 model were retrained while the other layers remained fixed, preserving the general features learned from the large-scale dataset and adding more specific features for the new dataset. The fine-tuned hybrid model, combining the strengths of CNN and EfficientNetB0, offered higher accuracy and generalization ability. The model delivered better results on the test data, and its performance was evaluated through classification reports and confusion matrices. Notably, the fine-tuned hybrid model achieved higher accuracy rates compared to the transfer learning model alone (*Pereira et al., 2016*).

## Quantum genetic algorithm

Quantum genetic algorithm (QGA) is an extension of classical genetic algorithms based on quantum mechanical principles. QGA aims to explore the solution space more effectively by using the concepts of quantum bits (qubits) and superposition. The basic components and working principles of this algorithm are as follows:

1. Beginning:

- The algorithm initializes the initial quantum state matrix $Q(t)$.
- By observing these situations, the initial population $P(t)$ is created.
- The population is evaluated and the best individual $B$ is kept.

2. Iterative process:

- The maximum number of generations $(t_{max})$ is selected.
- The number of iterations is increased $t = t + 1$
- A new population $P(t)$ is created by observing the states of the previous quantum state matrix $Q(t-1))$.
- The new population $P(t)$ is evaluated.
- The quantum state matrix $Q(t)$ is updated.
- The best individual in the new population is kept.

3. Update and result:

QGA aims to improve the overall performance of the population by updating qubits based on the best individual.

When the maximum number of generations is reached or other stopping criteria are met, the algorithm is terminated and the best individual *B* is given as the result.

The features and advantages of QGA are as follows. Quantum bits (Qubits), unlike classical bits, have superposition states that can take both values at the same time, not just '0' or '1'. A qubit state $|\psi\rangle$ is expressed as:

$$|\Psi> = \alpha|\theta> + \beta|1> \tag{1}$$

Here α and β are complex numbers, and $\alpha^2$ and $\beta^2$ values indicate the probabilities of the qubit being in the '0' and '1' states, respectively. As normalization condition:

$$|\alpha|^2 + |\beta|^2 = 1 \tag{2}$$

In superposition and parallel computing, the superposition state of qubits allows $2^m$ states to be represented simultaneously in m-qubit systems. This enables parallel exploration of the solution space and reduces computational complexity.

Regarding the global search capability, QGA reduces the possibility of getting stuck in local minima when searching for global optimal solutions. This allows the algorithm to find optimal solutions more effectively and quickly. In its effectiveness in feature selection, QGA enables the model to work more efficiently and accurately by identifying the most important features of the data when selecting features.

This is a significant advantage, especially in high-dimensional datasets. The QGA pseudo code is as follows.

```
Begin
t = 0;
Initialize Q(t); // Initialize quantum state matrix
Make P(t) by observing Q(t) states; // Create the initial population by observing Q(t) states
Evaluate P(t); // Evaluate the population
Choose a maximum number of generations tmax; // Select maximum number of generations
Store the best individual B among P(t); // Store the best individual among P(t)
While (not satisfied and t < tmax) Begin
t = t + 1;
Make P(t) by observing Q(t-1) states; // Create the new population by observing Q(t-1) states
Evaluate P(t); // Evaluate the new population
Update Q(t); // Update quantum state matrix
Store the best individual B among P(t); // Store the best individual among P(t)
end
end
```

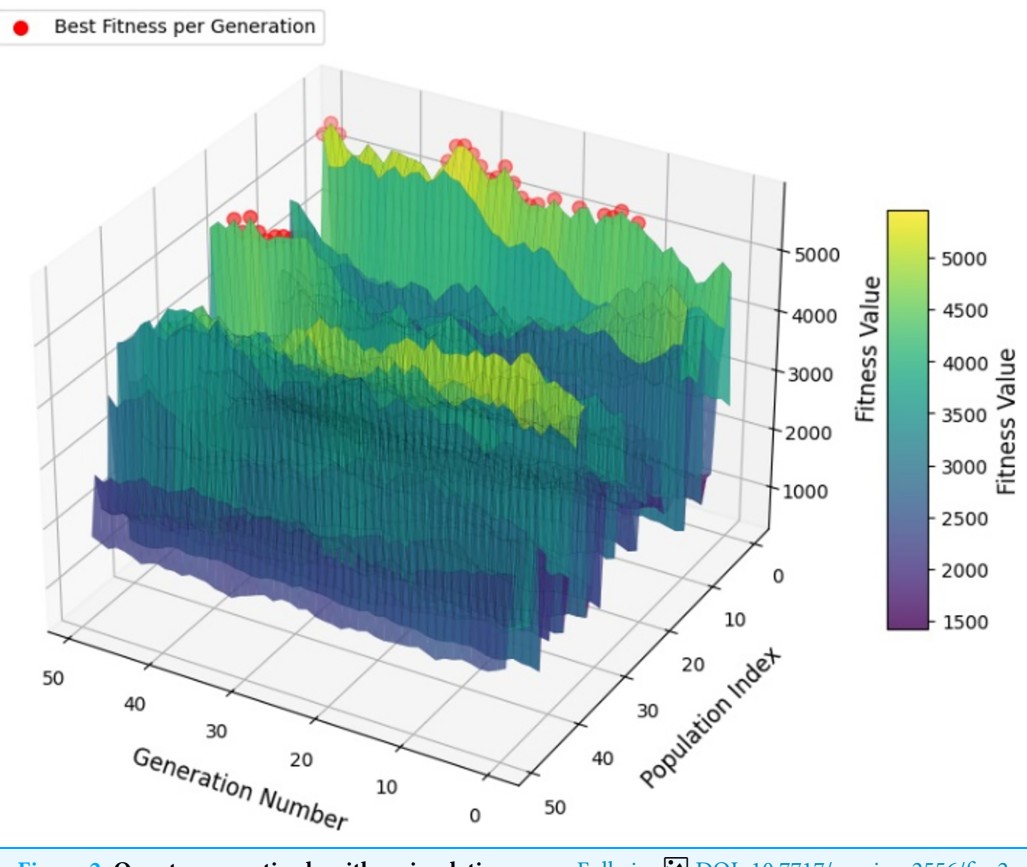

**Figure 2  Quantum genetic algorithm simulation.** 

In the update process, Qubits are updated to approximate the best individual angularly. The update formula is as follows:

$$\begin{bmatrix} \alpha_t \\ \beta_t \end{bmatrix} = \begin{bmatrix} cos(\Delta\theta_t) & -sin(\Delta\theta_t) \\ sin(\Delta\theta_t) & cos(\Delta\theta_t) \end{bmatrix} \begin{bmatrix} \alpha_{t-1} \\ \beta_{t-1} \end{bmatrix} \tag{3}$$

Here, the $\theta$ angle is best determined according to the fitness function of the individual (*Malossini, Blanzieri & Calarco, 2008*). A variant of QGA is given in Fig. 2.

## Proposed method

In this study, the EfficientNetB0 model was trained on image data. This model was used to solve the brain tumor classification problem and its accuracy was evaluated. Features were extracted from the trained EfficientNetB0 model. This process represents the features obtained from the last layers of the model. Feature extraction was performed on both training and test data. Feature selection was made using QGA on the features extracted from the EfficientNetB0 model. QGA was used to select the most important features in the data. A new model was created with features selected by QGA. This model was trained only on selected features. The new model trained with the selected features was re-evaluated on the test data. In summary, feature selection with QGA was applied on the features

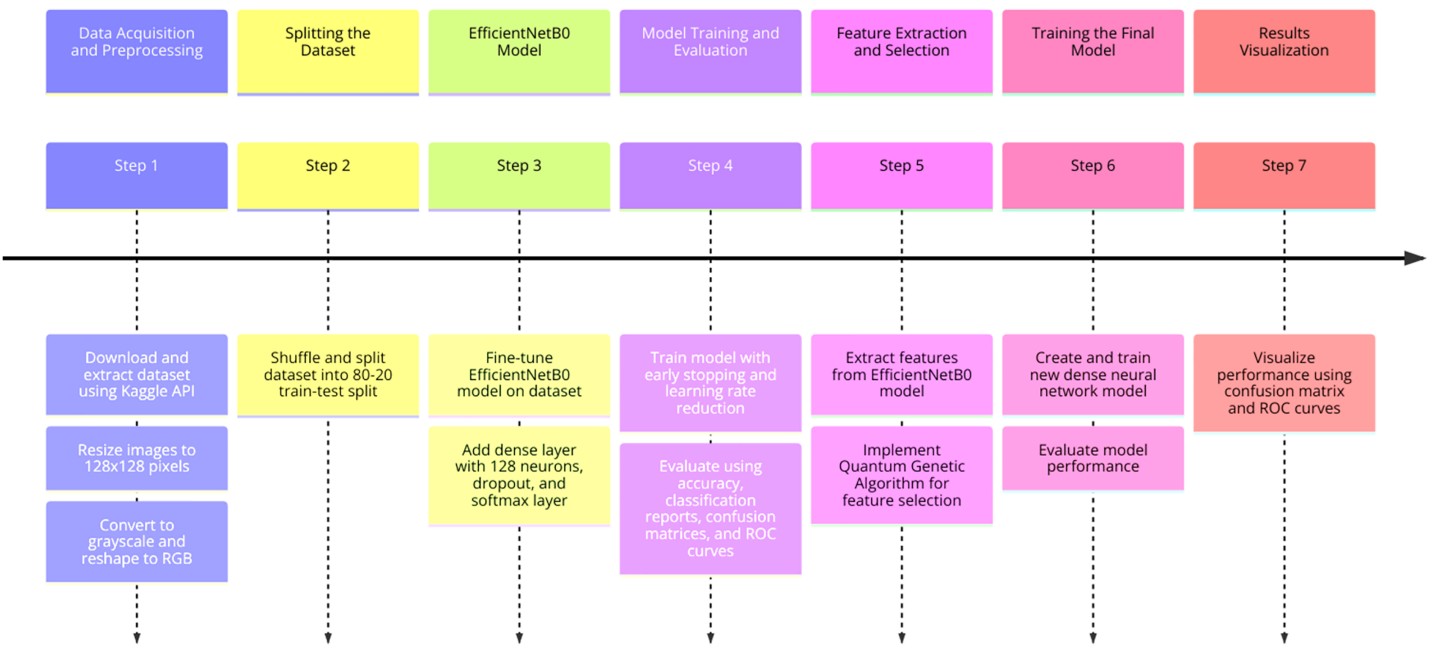

**Figure 3 Proposed approach for brain tumor classification.**

extracted from the EfficientNetB0 model, and it was aimed to make a more optimized classification by training a new model with these features. In this section, the step-by-step processes are given as follows and in Fig. 3. Brain tumor classification is a critical task for medical imaging and early diagnosis, and while many studies have explored deep learning approaches, there is still room for improvement in terms of accuracy and efficiency. This is especially important when dealing with limited and imbalanced datasets, where traditional methods may struggle to generalize well. The motivation behind our study is to enhance the performance of brain tumor classification systems by proposing a novel hybrid approach that combines the deep feature extraction capabilities of the EfficientNetB0 model with the powerful feature selection process of a QGA. This combination addresses the challenges of feature selection, improves the model's efficiency, and leads to better performance in classification, particularly in real-world clinical settings where accurate and rapid diagnosis is crucial. The unique contribution of this article lies in the integration of EfficientNetB0 with QGA-based feature selection (QGA-FS). No previous research has combined these two advanced techniques to address the brain tumor classification problem. This integration leverages EfficientNetB0's powerful feature extraction capabilities and QGA's ability to select the most important features, resulting in a more efficient and accurate model. The proposed method demonstrates superior accuracy compared to existing approaches. In our study, the hybrid model achieved accuracy rates of 98.36% on the Figshare dataset and 98.25% on the MRI dataset, outperforming other deep learning models and feature selection techniques in the literature. The proposed model was validated using two different datasets, emphasizing its generalization capacity

across different data types. This differentiates our approach from other studies that focus on a single dataset, making our solution more robust for real-world applications. While other methods use genetic algorithms for feature selection, the use of QGAs in combination with deep learning models for brain tumor classification is innovative. QGA enhances the model's ability to find optimal features in high-dimensional datasets, reducing computational costs and increasing classification accuracy. These contributions distinguish our article from existing studies, offering both innovation and practical value in improving the performance of brain tumor classification.

# EXPERIMENTS

## Datasets

The brain tumor dataset on the Kaggle platform was chosen due to its widespread use and its inclusion in many recent studies. The dataset provides a rich collection of MRI images and enables performance comparisons of commonly used models in the literature. In addition, the labeling quality, diversity, and easy accessibility of the dataset provide significant advantages for brain tumor classification studies. Although other datasets have been evaluated, this dataset was found to be more suitable due to factors such as being widely tested on different models and being constantly updated with feedback from the community. Therefore, we aimed to compare it with other studies in the literature by focusing on the Kaggle dataset in our study and analyze the success level of the proposed model in a wider reference framework. Two different datasets were used: the Figshare Brain Tumor Dataset and the MRI Brain Tumor Dataset. The Figshare Brain Tumor Dataset consists of images from three different tumor classes (glioma, meningioma, and pituitary tumor). MRI Brain Tumor Dataset consists of MRI images containing various types of tumors.

The first data set used in this study was collected by Tianjin Medical University between 2005 and 2010 and published online by *Jun (2017)*. The Figshare dataset consists of 3064 T1-weighted contrast-enhanced (CE) MRI images of glioma, meningioma and pituitary tumors. These images were obtained in three different planes (coronal, sagittal and axial) from 233 patients. Images are $512 \times 512$ pixels in size, with each pixel measuring 49 mm × 49 mm. The second data set was obtained from the Kaggle website. This dataset also includes Normal samples as well as glioma, meningioma and pituitary tumors. The Kaggle dataset is basically divided into two parts: training and testing samples. The images were collected from various web sources and published online after verification by a medical expert (*Bhuvaji et al., 2020*).

## Experimental protocol

The preprocessing steps applied to the datasets are as follows: All images were resized to $128 \times 128$ pixels. Black-and-white images were converted to RGB format and made three-channel. Pixel values were normalized to the range [0,1]. The datasets were split into 80% training and 20% testing. Two different approaches were tested using the EfficientNetB0 model and a CNN-based model: A model pre-trained on ImageNet was fine-tuned on

brain tumor images using transfer learning. Feature extraction and classification tasks were optimized using a CNN model trained from scratch. EfficientNetB0 and CNN were combined, where EfficientNetB0 was used for feature extraction and CNN for classification. During the parameter tuning process, the model hyperparameters were manually tested. The early stopping mechanism was used to prevent overfitting by stopping training early if no improvement was observed. Additionally, ReduceLROnPlateau was applied to reduce the learning rate when no improvement in validation loss was detected. The dropout rate used in the model was set to 0.5, which is a commonly accepted value in the literature. For the QGA-FS algorithm, the population size and maximum number of iterations were carefully tuned, and the mutation rate was set to 1%. To assess the robustness and generalization capability of the model, an 80–20 split ratio was used for training and testing. The data were randomly divided into training and testing sets, minimizing the risk of overfitting. Moreover, cross-validation was used to further improve the generalization ability of each model. A fixed random seed was employed in each experiment to ensure reproducibility of the results. The hyperparameters used during model training included a learning rate of 0.001, batch size of 32, epoch count of 50, early stopping (stopping training if no improvement in validation loss occurred for five epochs), and ReduceLROnPlateau (reducing the learning rate if no improvement occurred for five epochs).

## Evaluation metrics

The proposed method was evaluated with precision, recall, accuracy, F1 score, Cohen's Kappa score and Matthews correlation coefficient. Precision is specified in Eq. (4), Precision is specified in Eq. (5), and Accuracy is specified in Eq. (6). In Eq. (7), F1 Score is used as the harmonic average of these metrics. To evaluate the performance of the classification models, Cohen's Kappa coefficient and Matthews correlation coefficient (MCC) statistical measures are provided in Eqs. (8) and (9). Cohen's Kappa coefficient is a measure that adjusts for chance agreement when assessing the agreement between two classifiers (Cohen, 1960). It is particularly useful in imbalanced datasets where classification accuracy can be misleading. MCC is a balanced measure used in binary classification problems, evaluating the overall performance of the classifier. It provides reliable results even when there is an imbalance between positive and negative classes (Matthews, 1975).

$$Presicion = \frac{TP}{TP + FP} \tag{4}$$

$$Recall = \frac{TP}{TP + FN} \tag{5}$$

$$Accuracy = \frac{TP + TN}{TP + TN + FP + FN} \tag{6}$$

$$F1\ Score = 2 * \frac{Precision \times Recall}{Precision + Recall} \tag{7}$$

$$k = \frac{p_0 - p_e}{1 - p_e} \qquad (8)$$

$$MCC = \frac{(TP \times TN) - (FP \times FN)}{\sqrt{(TP + FP)(TP + FN)(TN + FP)(TN + FN)}} \qquad (9)$$

Here, TP represents true positives, FP represents false positives, FN represents false negatives, TN represents true negatives. F1 score is the harmonic mean of precision and sensitivity and is used to evaluate model performance on unbalanced data sets.

# RESULTS

## Proposed method results

In this study, first, the CNN method was used, as seen in Table 1. This method had precision values of 87%, 92% and 98% in the classification of meningioma, glioma and pituitary tumors, respectively. The overall accuracy rate was calculated as 92.65%. The transfer learning method had precision values of 94%, 98% and 98% in the classification of meningioma, glioma and pituitary tumors, respectively. The overall accuracy rate of this method was calculated as 97.22%. The hybrid model (CNN + EfficientNetB0) had precision values of 97%, 98% and 99% in classifying meningioma, glioma and pituitary tumors, respectively. The overall accuracy rate was calculated as 98.20%. The fine-tuned hybrid model (Fine Tuning Hybrid) had precision values of 50%, 67% and 69% in the classification of meningioma, glioma and pituitary tumors, respectively. The overall accuracy rate was calculated as 66.06%. Finally, the proposed EfficientNetB0 model and the QGA-FS method had precision values of 96%, 100% and 98%, respectively, in the classification of meningioma, glioma and pituitary tumors. The overall accuracy rate was calculated as 98.36%. Both Cohen's Kappa and MCC scores reveal that the proposed EfficientNetB0 model with QGA-FS outperforms the other methods, providing a highly accurate and reliable classification of brain tumors. The Hybrid (CNN + EfficientNetB0) model also performs well, while the Fine-Tuning Hybrid struggles significantly, as reflected in both Kappa and MCC scores.

In the second data set in Table 2, the CNN method was considered first. This method had precision values of 79%, 84%, 88% and 81% in the classification of no tumor, glioma tumor, pituitary tumor and meningioma tumor classes, respectively. The overall accuracy rate was calculated as 83.97%. The EfficientNetB0 model had precision values of 99%, 98%, 99% and 94% in the classification of no tumor, glioma tumor, pituitary tumor and meningioma tumor classes, respectively. The overall accuracy rate of this method was calculated as 96.86%. The hybrid model (CNN + EfficientNetB0) had precision values of 86%, 97%, 99% and 86% in the classification of no tumor, glioma tumor, pituitary tumor and meningioma tumor classes, respectively. The overall accuracy rate was calculated as 92.50%. The fine-tuned hybrid model (Fine Tuning Hybrid) had precision values of 93%, 96%, 96% and 98% in the classification of no tumor, glioma tumor, pituitary tumor and meningioma tumor classes, respectively. The overall accuracy rate was calculated as 95.64%. Finally, the proposed EfficientNetB0 model and the QGA-FS method had

**Table 1 Performance comparison of five different methods on the Figshare dataset.**

| Methods | Class | Precision | Recall | F1-score | Cohen's Kappa score | MCC |
|---|---|---|---|---|---|---|
| CNN | Meningioma | 0.87 | 0.84 | 0.85 | | |
| | Glioma | 0.92 | 0.94 | 0.93 | | |
| | Pituitary | | | | 0.886 | 0.886 |
| | Tumor | 0.98 | 0.97 | 0.97 | | |
| | Accuracy | | | 0.9265 | | |
| Transfer | Meningioma | 0.94 | 0.94 | 0.94 | | |
| | Glioma | 0.98 | 0.98 | 0.98 | | |
| | Pituitary | | | | 0.957 | 0.957 |
| | Tumor | 0.98 | 0.98 | 0.98 | | |
| | Accuracy | | | 0.9722 | | |
| (Hybrid) CNN+EfficientNetB0 | Meningioma | 0.97 | 0.95 | 0.96 | | |
| | Glioma | 0.98 | 1 | 0.99 | | |
| | Pituitary | | | | 0.972 | 0.972 |
| | Tumor | 0.99 | 0.98 | 0.99 | | |
| | Accuracy | | | 0.9820 | | |
| Fine Tuning Hybrid | Meningioma | 0.5 | 0.22 | 0.31 | | |
| | Glioma | 0.67 | 0.75 | 0.71 | | |
| | Pituitary | | | | 0.460 | 0.471 |
| | Tumor | 0.69 | 0.86 | 0.76 | | |
| | Accuracy | | | 0.6606 | | |
| Proposed EfficientNetB0 Model with QGA-FS | Meningioma | 0.96 | 0.97 | 0.97 | | |
| | Glioma | 1 | 0.99 | 0.99 | 0.975 | 0.975 |
| | Pituitary | | | | | |
| | Tumor | 0.98 | 0.98 | 0.98 | | |
| | Accuracy | | | 0.9836 | | |

precision values of 96%, 99%, 98% and 99%, respectively, in the classification of no tumor, glioma tumor, pituitary tumor and meningioma tumor classes. The overall accuracy rate was calculated as 98.25%. In summary, the Cohen's Kappa score and MCC both confirm that the proposed EfficientNetB0 with QGA-FS model provides superior performance with strong classification reliability and accuracy compared to other models, highlighting its potential for clinical applications. In general, the EfficientNetB0 model and QGA-FS method proposed for brain tumor classification have the highest accuracy rate compared to other methods and stand out as the most effective and reliable solution in distinguishing tumor types. This shows that the proposed method exhibits superior performance in automatic classification of brain tumors and can make a significant contribution in clinical applications.

Figure 4 shows the comparison of accuracy and loss values of different brain tumor classification models of the first data set throughout the training process. With the

**Table 2 Performance comparison of five different methods on MRI dataset.**

| Methods | Class | Precision | Recall | F1-score | Cohen's Kappa score | MCC |
|---|---|---|---|---|---|---|
| CNN | No_tumor | 0.79 | 0.74 | 0.77 | | |
| | Glioma_tumor | 0.84 | 0.81 | 0.83 | 0.813 | 0.813 |
| | Pituitary_tumor | 0.88 | 0.96 | 0.92 | | |
| | Meningioma_tumor | 0.81 | 0.79 | 0.80 | | |
| | Accuracy | | | 0.8397 | | |
| EfficientNetB0 | No_tumor | 0.99 | 0.96 | 0.97 | | |
| | Glioma_tumor | 0.98 | 0.95 | 0.97 | | |
| | Pituitary_tumor | 0.99 | 0.99 | 0.99 | 0.956 | 0.957 |
| | Meningioma_tumor | 0.94 | 0.97 | 0.95 | | |
| | Accuracy | | | 0.9686 | | |
| (Hybrid) CNN+EfficientNetB0 | No_tumor | 0.86 | 0.96 | 0.91 | | |
| | Glioma_tumor | 0.97 | 0.97 | 0.97 | | |
| | Pituitary_tumor | 0.99 | 0.85 | 0.91 | 0.904 | 0.906 |
| | Meningioma_tumor | 0.86 | 0.95 | 0.90 | | |
| | Accuracy | | | 0.9250 | | |
| Fine Tuning Hybrid | No_tumor | 0.93 | 0.96 | 0.95 | | |
| | Glioma_tumor | 0.96 | 0.98 | 0.97 | | |
| | Pituitary_tumor | 0.96 | 0.98 | 0.97 | 0.959 | 0.959 |
| | Meningioma_tumor | 0.98 | 0.93 | 0.96 | | |
| | Accuracy | | | 0.9564 | | |
| EfficientNetB0 Model with QGA-FS | No_tumor | 0.96 | 1.00 | 0.98 | | |
| | Glioma_tumor | 0.99 | 0.99 | 0.99 | | |
| | Pituitary_tumor | 0.98 | 0.99 | 0.99 | 0.976 | 0.976 |
| | Meningioma_tumor | 0.99 | 0.98 | 0.98 | | |
| | Accuracy | | | 0.9825 | | |

proposed EfficientNetB0 model, QGA-FS achieved the highest accuracy and lowest loss values, offering the best performance among all models.

Figure 5 compares the confusion matrices and receiver operating characteristic (ROC) curves of brain tumor classification models. QGA-FS with the proposed EfficientNetB0 model has the highest accuracy and discrimination ability for all classes and shows the best results in the ROC curve. This reveals that the proposed model offers the best performance and performs more accurate classification than other methods.

Figure 6 compares the accuracy and loss curves of brain tumor classification models for the second data set. With the proposed EfficientNetB0 model, QGA-FS achieved the highest level of accuracy and the lowest levels of loss values, demonstrating the best performance among all models. These results show that the proposed model is the most effective and reliable solution for brain tumor classification.

Figure 7 compares the confusion matrices and ROC curves of the brain tumor classification models in the second data set. With the proposed EfficientNetB0 model,

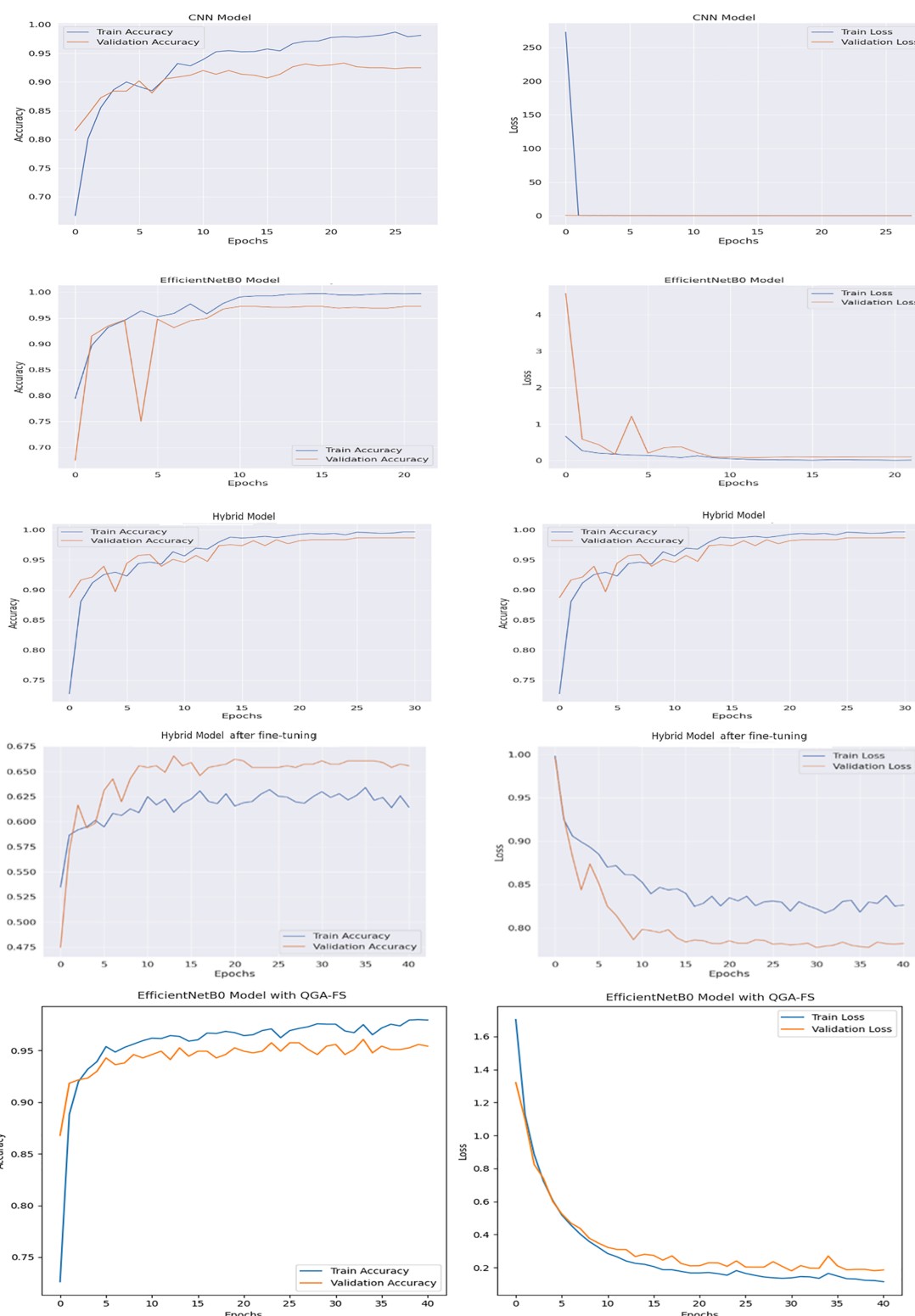

**Figure 4 Comparison of model accuracy and loss across epochs in the Figshare dataset, featuring CNN, EfficientNetB0, hybrid, fine hybrid, and the proposed EfficientNetB0 model with QGA-FS.**

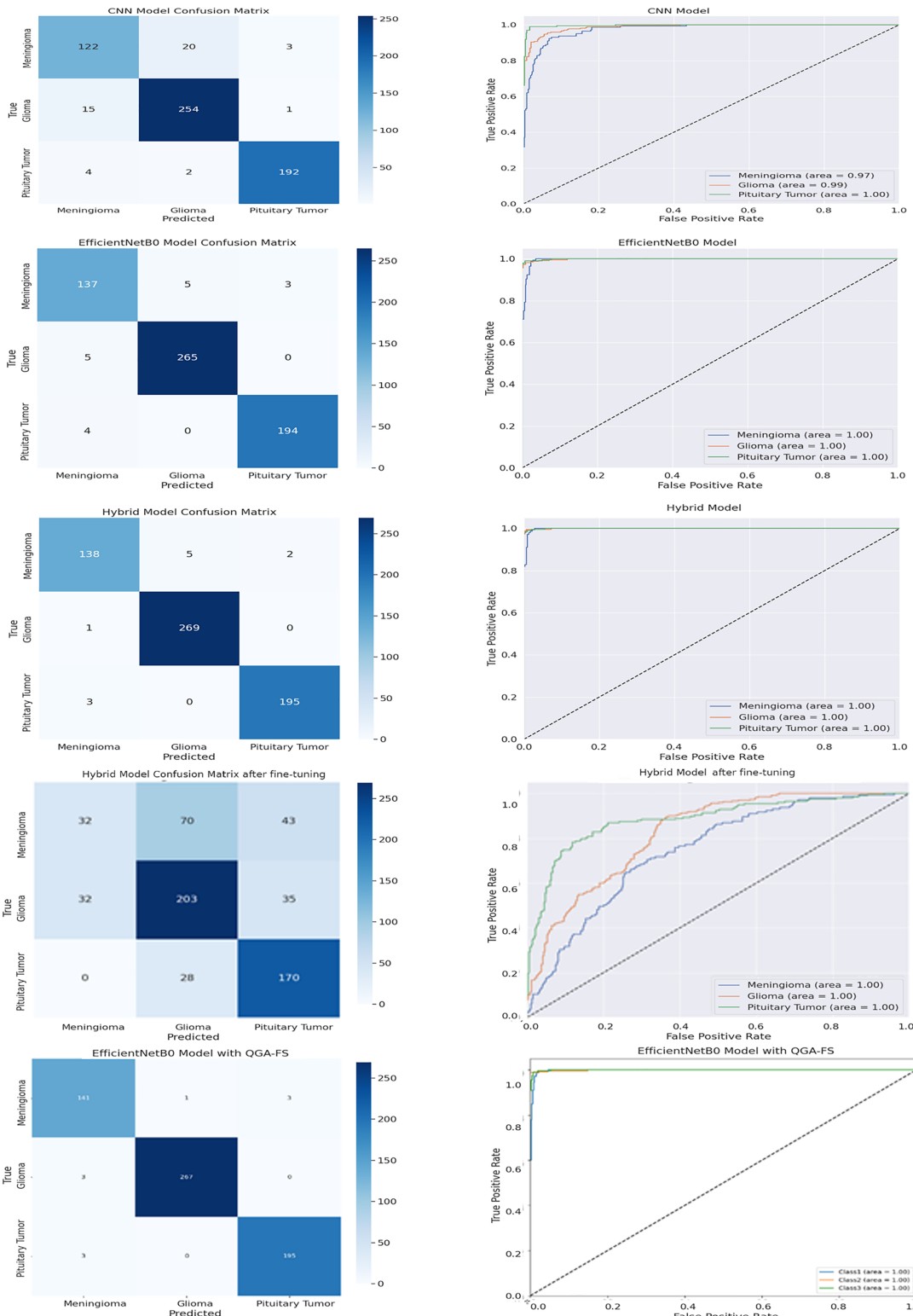

**Figure 5** Comparison of confusion matrices and ROC curves in the Figshare data set, featuring CNN, EfficientNetB0, hybrid, fine hybrid, and the proposed EfficientNetB0 model with QGA.

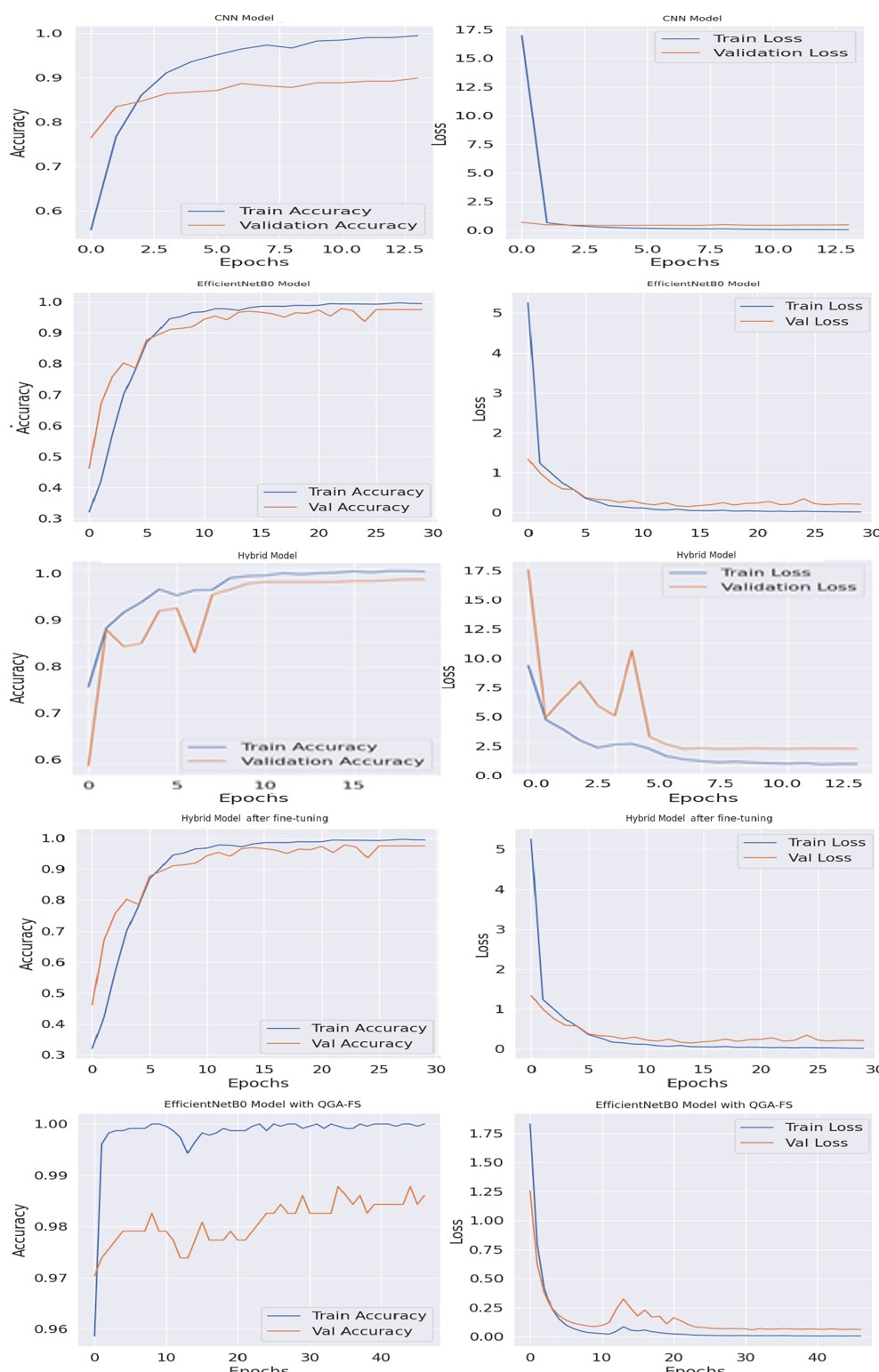

**Figure 6 Comparison of model accuracy and loss across epochs in the MRI dataset, featuring CNN, EfficientNetB0, hybrid, fine hybrid, and the proposed EfficientNetB0 model with QGA-FS.**

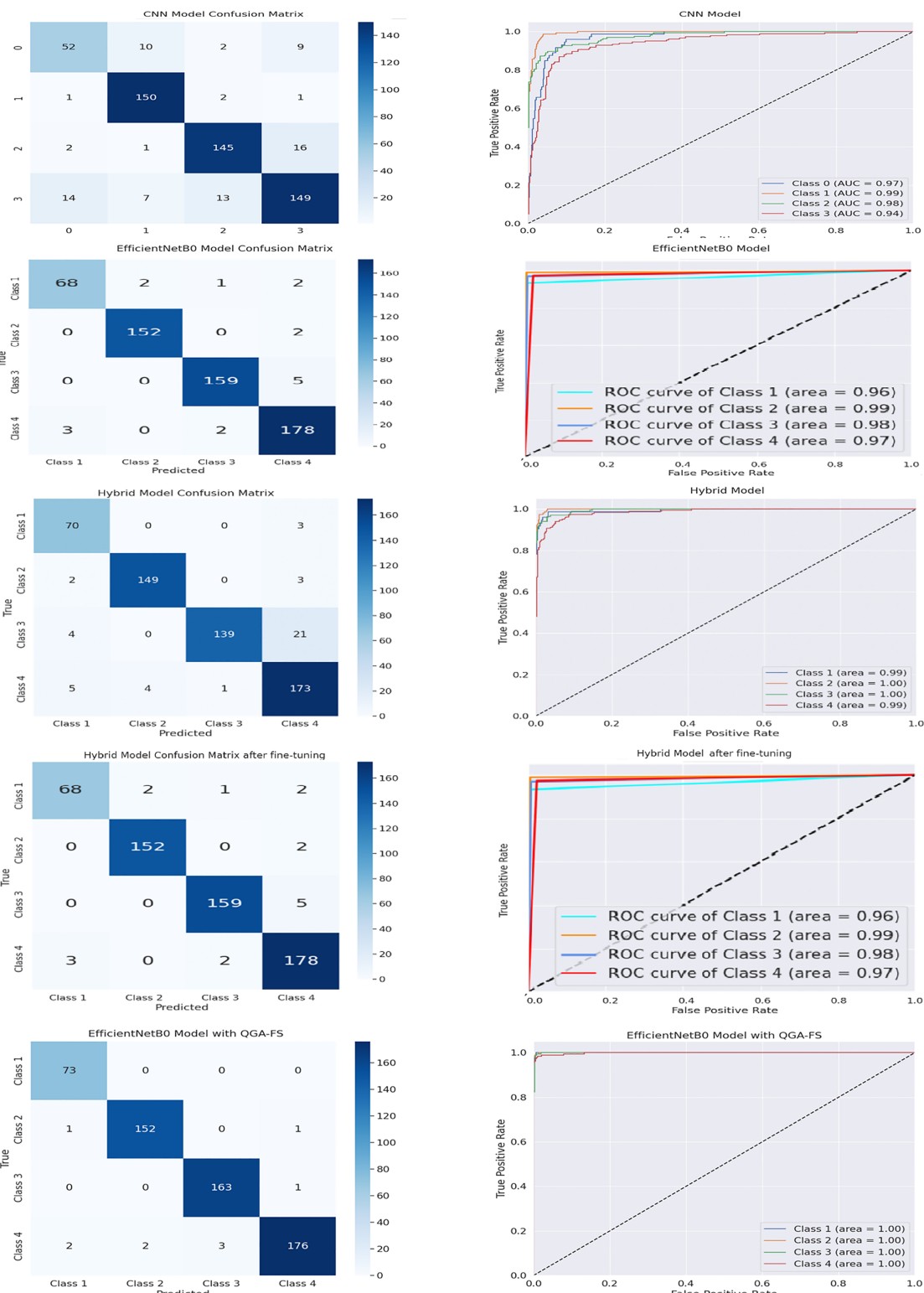

**Figure 7** Comparison of confusion matrices and Roc curves in the MRI dataset, featuring CNN, EfficientNetB0, hybrid, fine hybrid, and the proposed EfficientNetB0 model with QGA-FS.

**Table 3  Summary of techniques compared to state-of-the-art technologies for brain tumor classification using the Figshare and MRI datasets.**

| References | Methods | Datasets | Originality | Pros | Cons | Results |
|---|---|---|---|---|---|---|
| *Tummala et al. (2022)* | Vision Transformers Ensembling | Figshare | Transformer-based ensemble | State-of-the-art performance with Transformers | High computational cost | 98.70% |
| *Shaik & Cherukuri (2022)* | Multilevel attention mechanism (MANet) | Figshare, BraTS'2018 | Multi-level attention mechanism | Robust across datasets | Slightly lower accuracy on BraTS'2018 | 96.51% (Figshare), 94.91% (BraTS'2018) |
| *Aloraini et al. (2023)* | Hybrid transformer-enhanced CNN (TECNN) | BraTS 2018, Figshare | Combines transformers and CNN | High accuracy with hybrid model | High resource requirements | 96.75% (BraTS 2018), 99.10% (Figshare) |
| *Shyamala & Brahmananda (2023)* | Optimized feature reduction and regression neural network | MRI | Relief-based feature selection | Effective feature reduction for performance | Lower accuracy compared to others | 94.70% |
| *Sahoo et al. (2023)* | Efficient segmentation and classification using deep learning | MRI | Simultaneous segmentation and classification | Efficient and accurate for MRI data | No transformer inclusion | 97.30% |
| *Deepak & Ameer (2021)* | CNN features and SVM | MRI | Combination of CNN features and SVM classifier | Simplicity and effectiveness | May not generalize well on large datasets | 95.82% |
| *Sultan, Salem & Al-Atabany (2019)* | Multi-classification using deep neural network | MRI | Early work on multi-classification | Simple and effective neural network | Lower accuracy than more recent methods | 96.13% |
| *Swati et al. (2019)* | Transfer learning and fine-tuning | MRI | Transfer learning approach | Effective transfer learning with fine-tuning | Lower accuracy | 89.90% |
| *Pundir & Rajeev Kumar (2021)* | Transfer learning | MRI | Standard transfer learning | Effective for MRI data | Not highly original | 91.80% |
| Proposed EfficientNetB0 Model with QGA-FS | EfficientNetB0 with Quantum Genetic Algorithm | Figshare, MRI | Hybrid model combining QGA with EfficientNetB0 | High accuracy and effective feature selection | Computational complexity | 98.36% (Figshare), 98.25% (MRI) |

QGA-FS has the highest accuracy rate and lowest error rate for all classes and shows near-perfect results in the ROC curve. This reveals that the proposed model offers the best performance and performs more accurate classification than other methods.

## Comparison with state-of-the-art deep learning based techniques

Table 3 compares various deep learning techniques used in brain tumor classification. The accuracy rates of the QGA-FS with the proposed EfficientNetB0 model were found to be quite high compared to other methods. Hybrid deep learning approaches developed by *Raza et al. (2022)* aim to increase classification accuracy by using multiple methods. However, these models often fail to achieve ideal performance due to dataset limitations or architectural complexity. For example, Raza et al. achieved an accuracy rate of 96.30%. In contrast, the EfficientNetB0 with QGA-FS model, which uses QGA for feature selection, achieved higher accuracy rates of 98.36% (Figshare) and 98.25% (MRI). This demonstrates

that our model achieves better performance without requiring more complex architectures, thanks to its high accuracy and efficient feature selection capability.

*Tummala et al. (2022)* and *Aloraini et al. (2023)* achieved high accuracy rates (98.70% and 99.10%, respectively) using transformer-based hybrid models. However, the main disadvantage of these models is their high computational cost and resource requirements. The proposed EfficientNetB0 model offers lower computational cost compared to transformer models while achieving competitive accuracy rates, thanks to QGA-based feature selection. This makes our model more suitable for practical use.

Studies such as *Swati et al. (2019)* and *Pundir & Rajeev Kumar (2021)* used transfer learning methods, performing classification on MRI data using transferred knowledge. However, the lack of originality and lower accuracy rates (89.90% and 91.80%) are significant drawbacks. In contrast, the proposed model combines EfficientNetB0's powerful feature extraction capacity with QGA, offering a more innovative and effective approach compared to these transfer learning models.

Models like *Shyamala & Brahmananda (2023)*, which use feature reduction techniques, improve performance on high-dimensional datasets but still show lower accuracy rates (94.70%). While this study used relief-based feature selection, the proposed QGA-based feature selection model provides a more effective feature selection and higher accuracy due to its broader search space. This demonstrates that our model performs better even on more complex datasets.

Compared to other methods in the table, the proposed EfficientNetB0 Model with QGA-FS stands out in terms of originality, high accuracy rates, and effective feature selection (*Shaik & Cherukuri, 2022*; *Malla, Sahu & Alutaibi, 2023*; *Sahoo et al., 2023*; *Deepak & Ameer, 2021*). By combining QGA-based feature selection with EfficientNetB0's strong feature extraction capabilities, we achieved the highest accuracy rates in brain tumor classification. Additionally, we provide a more efficient and faster model by avoiding the high computational costs of transformer-based models. These results show the potential for our model to be used in clinical applications.

## DISCUSSION AND CONCLUSION

In our study, it was observed that CNN demonstrated strong performance in the first dataset, successfully distinguishing tumor types. Transfer learning enhanced classification performance, particularly providing very high accuracy in the classification of glioma and pituitary tumors. The hybrid model further improved classification performance by combining the strengths of the CNN and EfficientNetB0 models. However, the fine-tuned hybrid model did not deliver the expected performance improvement and actually reduced the model's performance. These results indicate that the proposed method achieved the highest accuracy among all methods and significantly improved feature selection and classification performance through QGA-FS.

Overall, the proposed EfficientNetB0 model and QGA-FS method for brain tumor classification reached the highest accuracy rate and demonstrated superior performance compared to other methods. This suggests that the proposed method offers an effective and reliable solution for the automatic classification of brain tumors.

In the second dataset, CNN successfully distinguished tumor types but provided lower accuracy in some classes (particularly in cases with no tumor and meningioma). The EfficientNetB0 model achieved very high accuracy and precision in tumor classification. The hybrid model improved classification performance by combining the strengths of both methods, but experienced performance drops in some classes (especially pituitary tumor). The fine-tuned hybrid model increased performance and achieved more balanced results. It was found that the proposed method achieved the highest accuracy among all methods and significantly enhanced feature selection and classification performance with QGA-FS.

This study investigates the effectiveness of combining deep learning with QGA in brain tumor classification. The proposed combination of the EfficientNetB0 model and QGA-FS achieved high accuracy rates in classifying brain tumors from MRI images. Specifically, accuracy rates of 98.36% were obtained on the Figshare dataset and 98.25% on the MRI dataset. These results demonstrate that the proposed approach offers superior performance compared to existing methods and can be used in clinical applications.

The main findings of the study show that when the EfficientNetB0 model's strong feature extraction capacity is combined with QGA's efficient feature selection, the model's classification accuracy and generalization ability are significantly increased. These results allow for faster and more accurate diagnostic processes by reducing dependency on manual feature extraction and classification methods in brain tumor diagnosis. Additionally, since the proposed model has lower dimensions and fewer learnable parameters, it reduces computational costs and processing times, providing real-world clinical results. This enhances the model's applicability in different scenarios.

The EfficientNetB0 model's strong feature extraction capacity stands out among deep learning models. The fact that this model is pre-trained on ImageNet allows the features learned from large datasets to be transferred to smaller and specific datasets, increasing classification accuracy. The QGA-FS method is more effective than traditional feature selection methods because quantum genetic algorithms offer a broader search space for feature selection and reduce the likelihood of getting stuck in local optima. This allows for the selection of better features that improve the model's overall performance and accuracy. QGA helps eliminate unnecessary or low-information features, particularly in high-dimensional datasets.

The findings of this study suggest that the proposed method can be used for the rapid and accurate diagnosis of brain tumors in clinical applications. The model's low dimensions and fewer learnable parameters reduce computational costs and processing times, increasing its applicability in real-world clinical scenarios. This can assist doctors in clinical settings by speeding up diagnostic processes and improving patient care.

This study contributes to future research by shedding light on the development of more efficient and effective methods for brain tumor classification. Future studies may focus on expanding the datasets, testing the model's applicability to different tumor types, and exploring the real-time use of the model in clinical settings. Additionally, integrations with other deep learning techniques and feature selection algorithms may be explored to further improve model performance.

Moreover, this study enhances the effectiveness of artificial intelligence and deep learning models in brain tumor classification, facilitating faster and more accurate early diagnosis. These advancements may enable healthcare policies to be reshaped. Specifically, the widespread adoption of AI-assisted diagnostic systems in healthcare systems can help improve patient care quality and reduce diagnostic costs. Furthermore, the widespread implementation of such systems may provide more automation in healthcare services, reduce the workload of doctors, and lead to more efficient patient management processes.

From an investor's perspective, this study demonstrates that the technological advancements in AI-based medical imaging present commercial opportunities. Such innovative models provide a strong foundation for new investments in healthcare technologies and offer attractive opportunities for projects at the intersection of big data, machine learning, and healthcare technologies. In particular, this model, which is suitable for clinical use and scalable, has the potential to create new business opportunities for medical device manufacturers and healthcare technology companies.

### Funding
The authors received no funding for this work.

### Competing Interests
The authors declare that they have no competing interests.

### Author Contributions
- Kerem Gencer conceived and designed the experiments, performed the experiments, analyzed the data, performed the computation work, authored or reviewed drafts of the article, and approved the final draft.
- Gülcan Gencer conceived and designed the experiments, performed the experiments, analyzed the data, performed the computation work, prepared figures and/or tables, authored or reviewed drafts of the article, and approved the final draft.

### Data Availability
The brain tumor dataset is available at figshare: Cheng, Jun (2017). brain tumor dataset. figshare. Dataset. https://doi.org/10.6084/m9.figshare.1512427.v5.

The Brain Tumor Classification (MRI) dataset is available at Kaggle: www.kaggle.com/sartajbhuvaji/brain-tumor-classification-mri.

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
