# Peer review of "Hybrid deep learning approach for brain tumor classification using EfficientNetB0 and novel quantum genetic algorithm"

_PeerJ Computer Science, doi:10.7717/peerj-cs.2556_

## Round 0.1 · original submission · Major Revisions

The review report from the expert reviewers are enclosed for your reference and revision of the manuscript.

Reviewer 1 ·

Basic reporting

All comments have been added in detail to the last section.

Experimental design

All comments have been added in detail to the last section.

Validity of the findings

All comments have been added in detail to the last section.

Additional comments

Review Report for PeerJ Computer Science
(Hybrid Deep Learning Approach For Brain Tumor Classification Using EfficientnetB0 And Novel Quantum Genetic Algorithm)

1. Within the scope of the study, an artificial intelligence-based hybrid system is proposed for the classification of brain tumors.

2. The importance of the subject and the main contributions of the study are clearly stated in the introduction section.

3. Although studies on brain tumor diagnosis related to artificial intelligence in the literature are mentioned in the Related works section, this section definitely needs to be detailed. In this context, it is recommended to add a literature table consisting of columns such as "dataset used, proposed method, originality, pros, cons, metrics, results" to this section and then to emphasize the difference of this study from the literature more when the studies in the literature are taken into consideration.

4. An open source dataset obtained from the Kaggle platform was preferred as the dataset in the study. Although there are other open source datasets related to brain tumors in the literature, it should be explained in more detail why this was chosen as the basis.

5. It is observed that EfficientNet was preferred as feature extraction. In the literature, especially in studies conducted on biomedical images and where hybrid models are developed, very different feature extractions can be used. For this reason, the reason for choosing this model should be emphasized more clearly.

6. The use of Quantum Genetic Algorithm in the hybrid model has definitely increased the originality and quality of the study.

7. Although the evaluation metrics seem to be sufficient at a basic level, it is recommended to obtain Cohen’s Kappa score and Matthews Correlation Coefficient score for a more accurate analysis and interpretation of the results.

In conclusion, the study has the potential to contribute to the literature, but attention should be paid to the above sections.

Reviewer 2 ·

Basic reporting

-There are already many research articles available on the similar topic what is the need of this paper? What is the motivation if this paper? What is the novel contribution of this article compared to other existing articles?
- The results should be discussed more thoroughly in light of the existing literature. This is a very important point to address.
- In the conclusion section, a more detailed discussion of the policy implications would make the paper richer and more informative for investors and policymakers.
- While the paper mentions extensive comparative experiments on various public datasets, it lacks specific details about the experimental setup, such as the selection criteria for datasets, parameter tuning procedures, and validation techniques. Providing this information would enhance the reproducibility of the results and allow readers to better understand the robustness of the proposed model
- The discussion was too shallow and did not explain why the proposed method was superior. The authors are also requested to focus on the obtained results and reflect the proposed method's effect on them. The robustness about the method have not been discussed.
- Check the mathematical notation, especially for the proposed method. This will facilitate the new readers' tracking and applying of the proposed method and get the same results.
- Various figures are not explained well. I suggest adding a brief description of each figure in their captions.
- A complete description of the experimental protocol is missing.
- There are many algorithm parameters in the proposed method. What's the influence of these parameters?
- I suggest that the authors introduce certain taxonomy, at least through subsections.

Experimental design

no comment

Validity of the findings

no comment

Additional comments

no comment

---

## Round 0.2 · accepted · Accept

Thanks for revising based on reviewers comments. Congratulations

Reviewer 1 ·

Basic reporting

All comments have been added in detail to the last section.

Experimental design

All comments have been added in detail to the last section.

Validity of the findings

All comments have been added in detail to the last section.

Additional comments

Review Report for PeerJ Computer Science
(Hybrid Deep Learning Approach For Brain Tumor Classification Using EfficientnetB0 And Novel Quantum Genetic Algorithm)

Thank you for the revision. Considering both the detailed responses to the comments and the contribution of the paper to the literature, I recommend that the paper be accepted. Best regards.

Reviewer 2 ·

Basic reporting

- I have reviewed the new version of the manuscript and consider that the authors have satisfactorily addressed the reviewers' comments. The manuscript has improved its quality. Also, its contribution to the state of the art is clear

Experimental design

OK

Validity of the findings

OK